



# Impacts of three types of solar geoengineering on the North Atlantic Meridional Overturning Circulation

Mengdie Xie[1], John C. Moore[1,2,3], Liyun Zhao[1,4], Michael Wolovick[1], Helene Muri[5]

[1] College of Global Change and Earth System Science, Beijing Normal University, Beijing, 100875, China

[2] CAS Center for Excellence in Tibetan Plateau Earth Sciences, Beijing, 100101, China

[3] Arctic Centre, University of Lapland, Rovaniemi, 96101, Finland

[4] Southern Marine Science and Engineering Guangdong Laboratory (Zhuhai), Zhuhai, 519082, China

[5] The Industrial Ecology Programme, Norwegian University of Science and Technology, Trondheim, Postboks 8900, NO-7491, Norway

*Correspondence to*: John C. Moore (john.moore.bnu@gmail.com)

**Abstract.** Climate models simulate lower rates of North Atlantic heat transport under greenhouse gas climates than at present due to a reduction in the strength of the North Atlantic meridional overturning circulation (AMOC). Solar geoengineering whereby surface temperatures are cooled by reduction of incoming shortwave radiation may be expected to ameliorate this effect. We investigate this using six Earth System Models running scenarios from GeoMIP (Geoengineering model

intercomparison project) in the cases of: i) reduction in the solar constant, mimicking dimming of the sun; ii) sulfate aerosol injection into the lower equatorial stratosphere; and iii) brightening of the ocean regions mimicking enhancing tropospheric cloud amounts. We find that despite across model differences, AMOC decreases are attributable to reduced air-ocean temperature differences, and reduced September Arctic sea ice extent, with no significant impact from changing surface winds or precipitation-evaporation. Reversing the surface freshening of the North Atlantic overturning regions caused by decreased

summer sea ice sea helps to promote AMOC. Comparing the geoengineering types after normalizing them for the differences in top of atmosphere radiative forcing, we find that solar dimming is more effective than either marine cloud brightening or stratospheric aerosol injection.

## 1 Introduction

Geoengineering, that is the deliberate and large-scale manipulation of the Earth's climate, has been proposed as a way to

mitigate or offset some of the impacts of anthropogenic global warming (Keith, 2000). Solar Radiation Management (SRM) is one of the fundamental geoengineering methodologies, increasing Earth's albedo to reduce the net solar irradiance reaching Earth, thus balancing longwave greenhouse gas (GHG) forcing (Niemeier et al., 2013). Stratospheric aerosol injection (SAI) whereby aerosols aloft reflect incoming solar radiation, and marine cloud brightening (MCB), that is introducing aerosols into the marine boundary layer and thereby increasing cloud droplet numbers and hence their reflectivity (Jones et al., 2011; Ahlm

et al., 2017) are the most commonly discussed methods. Another hypothesized method of SRM is simply blocking some incoming solar radiation before it reaches the Earth (Angel, 2006), known as solar dimming or sunshade geoengineering, has

proven useful because of the climate response insights it provides. All three methods can cool global mean temperatures, but the tropospheric marine injection in MCB produces greater disparity in regional climate effects, such as on precipitation (Muri et al., 2018; Kravitz et al., 2018). This is not necessarily an inherent disadvantage relative to SAI since it is plausible that

combining different SRM methods may deal with regionally-specific deleterious impacts of climate change better than any one method alone (Cao et al., 2017).

The most comprehensive model simulations of climate under SRM scenarios to date come from the GeoMIP (Geoengineering Model Intercomparison Project; Kravitz et al., 2013; 2016). These experiments are highly idealized – for example, offsetting

of a sudden quadrupling of $CO_2$ concentrations by turning down the solar constant. The point of the experiments is to examine the mechanistic behavior of the climate system when subjected to different styles of SRM forcing in comparison with pure greenhouse gas (GHG) forcing. The global nature of the scenarios allows for sufficient signal/noise ratio to discern impacts on various parts of the climate system in a reasonable simulation period with Earth System Models (ESM). There are still technical barriers and risks to doing both MCB (Latham et al., 2012), and SAI (Smith and Wagner, 2018), while doing sunshade

SRM is well beyond the bounds of likelihood (Angel, 2006). We are not advocating implementation any time soon. Instead, our aim with this paper is to use the GeoMIP experiments to investigate the mechanistic effect that SRM, and MCB, has on an important and unique climate sub-system: the Atlantic Meridional Overturning Circulation (AMOC).

The AMOC describes an ocean circulation that is highly correlated with the poleward transport of heat in the sub-tropical

North Atlantic (Johns et al., 2011). AMOC transports 90% of the ocean meridional heat transport at 26.5° N (Johns et al., 2011). The upper branch of AMOC transports warm surface fresh water from the tropics northwards where it loses heat, densifies, and eventually descends in the North Atlantic deep convection regions. AMOC releases about 1.25 PW of heat from the sea to the atmosphere between 26°N and 50°N which warms the North Atlantic region and northern Europe, while the deep branch transports cold salty deep water southward that ultimately fills a large fraction of the global ocean basins (Buckley and

Marshall, 2016; Chen and Tung, 2018;). AMOC is mainly driven by global density gradients due to surface heat and freshwater fluxes (more details are available in for example, McCarthy et al., 2019). Its potential for net northward heat transport is unique and plays an essential role in global climate and the redistribution of heat. Changes to the heat and salt fluxes carried by AMOC must produce various climatic effects, such as changes in tropical cyclone number and intensity, and hence hurricanes impacting its western boundaries, and changes in monsoonal rainfall in Africa and India (Buckley and Marshall, 2016).

Therefore, any side effects that SRM may have on AMOC has the potential to produce wide-ranging, societally relevant consequences.

It has proven very difficult to observe the magnitude of AMOC directly (McCarthy et al., 2019; Send et al., 2011), so the observational evidence for AMOC strength remains limited. It has been possible to accurately quantify the temporal variation

of AMOC only since April 2004 when continuous observations of AMOC began at 26.5°N by the Rapid Climate Change–





Meridional Overturning Circulation and Heat flux Array–Western Boundary Time Series (RAPID–MOCHA–WBTS) project in the North Atlantic (Smeed et al., 2018). The mean strength of AMOC from April 2004 to February 2017 was 17.0 Sv with a standard deviation of 4.4 Sv (Frajka-Williams et al., 2019). The 26.5° N array observations provide information on the short term inter-annual and seasonal variability of AMOC. Annually AMOC ranges in strength from 4 to 35 SV and also has seasonal

characteristics (Frajka-Williams et al., 2019). AMOC intensity decreased significantly during 2004-2012 but was then statistically unchanged between 2012 and 2017 (Smeed et al., 2018). The decline is thought to be related to the Atlantic Multidecadal Oscillation, and not to the long-term external climate forcing. The less than two-decade observational record is insufficient to detect the effect of external climate stress on AMOC (Roberts et al., 2014). Numerical climate models show a slight decline of AMOC in the historical period and predict that AMOC will continue to weaken in the 21st century (Cheng et

al., 2013). Predicted AMOC decline is stronger in more recent models than in earlier ones, with modern ensemble mean estimates suggesting declines between 6 and 8 Sv (34–45%) by 2100 (Weijer et al., 2020). Compared with the past 1500 years, AMOC has experienced an exceptional weakening in the past 150 years (Thornalley et al., 2018).

The external forcing factors that control AMOC intensity depend on the time scale being considered. On short time scales

(monthly to seasonal), change in wind stress can be the main factor affecting its intensity (Zhao and Johns, 2014), but on long time scales (interannual to interdecadal) the seawater density affected by fresh water flux and sea-air heat flux are the main factors (Smeed et al., 2018).

To date, little research on the oceanic response at high northern latitudes under SRM has been published (Malik et al., 2020;

Muri et al., 2018; Smyth et al., 2017). Some research has been done on AMOC under sunshade geoengineering (Hong et al., 2017) and under SAI (Muri et al., 2018; Moore et al., 2019; Tilmes et al., 2020). As with GHG forcing alone, these studies found a weakening of AMOC relative to present day under sunshade geoengineering, mainly in response to the change of heat flux in the North Atlantic, with little influence from the changes of freshwater flux and wind stress (Hong et al., 2017). However, the AMOC is less weakened under sunshade geoengineering than with GHG forcing alone (Hong et al., 2017). Under SAI

experiments, AMOC declines seen under greenhouse forcing are consistently reversed (Moore et al., 2019; Tilmes et al., 2020; Muri et al., 2018). All ESM simulation results agree that SAI mitigates weakening of the AMOC as compared with the GHG control experiments. Hence AMOC is closer to the present-day with sunshade and SAI SRM than without, but very little research on AMOC under MCB experiments has yet been published (Muri et al., 2018).

Here, we evaluate and compare the potential for MCB to offset changes under GHG forcing to AMOC and its effectiveness and mechanistic behavior relative to SAI and sunshade geoengineering based on the same 6 ESM, (Table 1). We focus on the response of northward ocean heat transport, freshwater flux, sea-air heat flux, the AMOC strength, atmospheric wind stresses and Arctic sea ice extent.



## 2 Data and Methods

We analyze monthly output from all ESM that participated in GeoMIP with sufficient data fields available (Table 1). The G1 and G1oceanAlbedo experiments are very idealized simulations where incoming solar radiation is reduced to balance the longwave radiative forcing of quadrupled $CO_2$ relative to pre-industrial concentrations. The G4 and G4cdnc experiments represent somewhat more real-world scenarios where the background greenhouse concentration rises as specified by the RCP4.5 scenario while SRM is prescribed either by constant amounts for SAI (G4) or increased cloud condensation nuclei over the ocean (G4cdnc; see Section 2.1 for more information and Kravitz et al., 2011; 2013, for a full description of the experiment design). Hence there are three control simulations: i) the standard piControl specifying pre-industrial conditions; ii) abrupt4×$CO_2$ specifying the standard abrupt quadrupling of $CO_2$; and iii) the RCP4.5 scenario specified under the Climate Model Intercomparison Project Phase 5 (CMIP5; Taylor et al., 2012). Not all the ESM we use have every simulated climate field that we would like; some lack heat and water flux data or sea ice extents (Table S1).

**Table 1: Earth System Models used in this study.**

| Model | Reference | Ocean component | Ocean Lat×Lon×Depth |
|---|---|---|---|
| BNU-ESM | Ji et al. (2014) | MOM4p1 (Griffies, 2010) | (1/3°~1°) ×1° ×L50 |
| CanESM2 | Yang et al. (2012) | NCAR CSM Ocean Model (Gent et al., 1998) | 0.94°×1.41°×L40 |
| HadGEM2-ES | Collins et al. (2011) | HadGEM2-O | (1/3°~1°) ×1°×L40 |
| ISPL-CM5A-LR | Dufresne et al. (2013) | NEMO | 1.875°×3.75°×L39 |
| MIROC-ESM | Watanabe et al. (2011) | COCO3.4 (K-1 model developers, 2004) | (0.5°~1.7°) ×1.4°×L44 |
| NorESM1-M | Bentsen et al. (2013) Iversen et al. (2012) | a developed version of MICOM | 1°×1°×L70 |

The response of the oceans is expected to be much slower than the atmosphere. Typically, in the sunshade experiments which invoke abrupt and strong forcing, the first decade of the simulations has not been included in the analysis to mitigate this issue. It is of course unlikely that the deep ocean would be close to a steady state within centuries of beginning geoengineering experiments, but to be practical we assume that the scenario responses after the first decade are sufficiently different from each other to explore impacts. Most GeoMIP scenarios run for 50 years, and while some GHG and control scenarios run longer, we limit the analysis of all scenarios to the same duration for statistical convenience. We test for significance at the 95% level using the non-parametric Wilcoxon signed rank test.



## 2.1 Experiments

120 Schematic representation of the experiments are shown in Fig. 1. G1oceanAlbedo is part of the Phase 2 GeoMIP experiments (Kravitz et al., 2013; 2015) and designed to mimic the G1 solar dimming experiment (Kravitz et al., 2011). Both are based on the CMIP5 abrupt4×$CO_2$ experiment and started from a stable pre-industrial climate run i.e., the CMIP5 experiment piControl (Taylor et al., 2012). In the G1 experiment, the radiative forcing from an abrupt quadrupling of $CO_2$ concentrations above preindustrial levels is offset by a uniform insolation reduction, thereby mimicking sunshade geoengineering. In

125 G1oceanAlbedo, the radiative forcing from abrupt4×$CO_2$ is instead compensated for using a uniform increase in albedo in the ESM ocean-covered grid cells (Fig. 1a). The G4 experiment, by contrast, starts with the RCP4.5 scenario as a baseline and then employs an injection rate of SAI (5 Tg of SO2 per year) into the equatorial lower stratosphere between the years 2020 and 2069 (Figure 1c). The G4cdnc scenario is similar, except that the stratospheric aerosols are replaced by a 50% increase in the cloud number droplet concentration in low clouds over the global ice-free oceans. In both G4 and G4cdnc, the amount of

130 geoengineering is held fixed over time, rather than being adjusted to balance the radiative forcing due to GHGs.



**Figure 1: Schematics of the four experiments outlined in this paper, based on Kravitz et al. (2011; 2013). (a) G1 is started from a preindustrial control run, longwave forcing (blue) from quadrupled GHG forcing is compensated by a fixed reduction in the solar constant (red) to leave net zero forcing (black), the experiment is for 50 years duration. (b) In G1ocean-albedo the equivalent balance is obtained by an increase in ocean albedo. (c) G4 is started from 2020 and ends in 2069 branching from RCP4.5 with 5 Tg yr$^{-1}$ SO$_2$ injected into the equatorial lower stratosphere. (d) In G4cdnc the shortwave forcing comes from a constant 50% increase in cloud droplet number concentration in oceanic low clouds.**

### 2.2 AMOC index

The AMOC index (Cheng et al., 2013) is defined as the annual-mean maximum volume of the transport stream function at 30°N in the North Atlantic (in Sverdrups (Sv)). The transport stream function is described by the integral of the meridional transport from the surface to the bottom depth at the given latitude (here 30°N):

$$\Psi(z, lat) = \int_{z}^{0} \int_{\lambda_E}^{\lambda_W} V \cos(lat) \, dx \, dz \,, \tag{1}$$





where Ψ is the overall transport stream function, z is the bottom depth, lat is latitude, λE and λW represent the eastern and western meridians respectively. V is the meridional ocean velocity.

## 2.3 Northward Heat Transport

In this study, we use the ocean potential temperature and the ocean meridional velocity to calculate the northward heat transport, H (Stouffer et al., 2017):

$$H(lat) = Cp \cdot \oint_{lat} \int_z^0 \rho \cdot T \cdot V dz \, d(lon) \, , \tag{2}$$

H(lat) is the ocean heat transport in the latitude, Cp is the ocean specific heat capacity, ρ is the ocean potential density, lon is longitude, and T is ocean potential temperature.

## 3 AMOC response and its impact

### 3.1 Experiments

**Table 2: Differences in average AMOC index, upward heat flux (W m⁻²), September sea ice extent (10⁶ km²), and top of atmosphere radiation (W m⁻²) over the 40-year analysis period. Bold entries denote differences significant at the 95% level in the Wilcoxon signed rank test. G1oa refers to G1oceanAlbedo, and PiC refers to piControl. Individual ESM results are shown in Tables S2-5.**

| Experiments | AMOC Flux (Sv) | Upward Heat flux (Wm⁻²) | Arctic September Sea Ice (10⁶ km²) | TOA radiation (Wm⁻²) |
|---|---|---|---|---|
| 4xCO₂-piC | **-6.0** | **-37.2** | **-5.9** | **2.7** |
| G1-piC | -0.7 | -8.3 | -0.3 | 0.1 |
| G1oa-piC | **-1.4** | **-17.7** | **-1.6** | -0.4 |
| G1oa-4xCO₂ | **4.6** | **24.3** | **4.2** | **-3.0** |
| G1-4xCO₂ | **5.3** | **28.9** | **5.6** | **-2.5** |
| G4cdnc-RCP4.5 | **1.3** | **5.4** | 1.4 | **-0.8** |
| G4-RCP4.5 | **0.9** | 2.7 | 1.0 | -0.3 |
| G1oa-G1 | -0.7 | **-6.7** | **-1.3** | -0.5 |
| G4cdnc-G4 | 0.6 | 2.6 | -0.2 | -0.4 |



**Figure 2: 11 year running annual means simulated by the 6 ESM, and the multi-model ensemble mean (black curve), of the AMOC strength (Sv) over the 40-year analysis period under (a) piControl, (b) abrupt4×CO₂ and (c) RCP4.5. The gray band in (a) is the range of AMOC intensity (17.0 ± 4.4 Sv) measured by the RAPID- MOCHA (Frajka-Williams et al., 2019). Panels (d-f) show AMOC anomalies (Sv) and panels (g-i) the percentage changes relative to the other scenarios: Left column (d,g) relative to piControl; Middle (e,h) relative to global warming scenarios (RCP4.5 and abrupt4xCO₂); Right (f,i) relative to other geoengineering scenarios (G1oa-**





**G1; G4cdnc-G4). Colored bands in panels (d-i) represent the across-ESM spread. G1oa refers to G1oceanAlbedo, and PiC refers to piControl.**

Under the piControl scenario, the six ESM ensemble mean AMOC index is about 17.9 Sv, which is consistent with the average AMOC strength (17.7±0.3 Sv), from the RAPID- MOCHA array (Weijer et al., 2020), (Fig. 2a). Under RCP 4.5, the AMOC intensity decreases by about 2.4 Sv from 2020 to 2069 (Fig. 2c; Table 2), consistent with previously published ESM simulation results (Cheng et al., 2013; Weijer et al., 2020; Muri et al., 2018). Compared with the piControl, the AMOC intensity in the 50th year of abrupt4×CO$_2$ and RCP 4.5, decreased by about 7.9 Sv (42%), compared with a 15% reduction under RCP4.5 (Fig. 2g). which is consistent with the lower GHG forcing under the RCP4.5.

Under G1 and G1oceanAlbedo scenarios, the average AMOC strength over the 40-year analysis period increased by about 5.3 Sv and 4.6 Sv relative to abrupt4×CO$_2$ (Table 2). Compared with abrupt4×CO$_2$, the AMOC intensity in the 50th year of G1 and G1oceanAlbedo, increased by about 7.2 Sv (41%) and 6.2 Sv (35%) (Fig. 2 e, h). The average AMOC intensity is weaker by 0.7 Sv and 1.4 Sv under G1 and G1oceanAlbedo (Table 2) than under piControl, differences which are significant at the 95% level. MIROC-ESM simulated a slightly stronger AMOC under G1oceanAlbedo than under G1 (Table S2), but the other five ESMs and the ensemble mean agree that AMOC under the G1 scenario is stronger than that under G1oceanAlbedo. Even though G1oceanAlbedo is designed to produce radiative forcing over ice-free oceans, it is significantly less effective at restoring AMOC to piControl levels than the global forcing applied under G1.

Both G4cdnc and G4 apply constant reductions to shortwave solar radiation, but in contrast with the abrupt4×CO$_2$ scenario, the GHG concentrations continue to rise in these scenarios as specified by RCP4.5. Under the G4 and G4cdnc scenarios, the average AMOC strength over the 40-year analysis period increased by about 0.9 Sv and 1.3 Sv relative to RCP4.5 (Table 2), both significantly different from RCP4.5 (Table 2). The difference between G4cdnc with G4 over the 40-year analysis period is also significant. Five ESM, and the ensemble mean agree that the ocean-only forcing under G4cdnc is more effective than the global G4 forcing for restoring AMOC to present-day strength (Table S2).

We may thus conclude that the four geoengineering experiments mitigate AMOC weakening caused by the forcing of GHG, but the mitigation efficacies are different. Generally, mitigation of AMOC weakening under G4cdnc is more than with G4, but weaker than G1 solar dimming. G1oceanAlbedo is more effective than G4cdnc, but these scenarios were not designed to have identical forcing, so we shall discuss their relative efficacy later in the Discussion.

## 3.2 Northward heat transport response

AMOC transports heat from low latitudes to high latitudes at the upper levels of the ocean. How will the northward heat transport change with the change of AMOC intensity under different styles of SRM?





**Figure 3: Meridional distribution of the average northward heat transport (PW) over the 40-year analysis period at Atlantic Ocean (depth 0-700 m) under (a) piControl, (b) abrupt4×CO₂, and (c) RCP4.5. Panels (d-f) show northward heat transport anomalies relative to the other scenarios: Left column (d) relative to piControl; Middle (e) relative to global warming scenarios; Right (f) relative to other geoengineering scenarios. Colored bands in panels (d-f) represent the across-ESM spread.**

Under piControl, the 6 ESMs ensemble mean northward heat transport at 26.5°N in the Atlantic Ocean is about 1.27 PW (Fig. 3a), which is consistent with Johns et al. (2011) estimate of 1.25 PW for meridional heat transport in the Atlantic Ocean from 2004 to 2007.

Under the two global warming scenarios (RCP4.5 and 4xCO₂), the northward heat transport at the Atlantic basin to the south of 60°N decreases significantly relative to piControl, particularly between 30°N and 50°N, and increases between 60°N and 70°N (Fig 3d). Under the abrupt4×CO₂ and RCP4.5 scenarios, AMOC weakening reduces the heat transported northward by about 0.51 PW and 0.07 PW between 30°N and 50°N relative to piControl.





Under G1 and G1oceanAlbedo scenarios, the northward heat transport increased by about 0.45 PW and 0.4 PW between 30°N and 50°N relative to abrupt4×CO$_2$ (Fig. 3e). The northward heat transport weakening between 30°N and 50°N caused by GHGs is significantly mitigated by G1 and G1oceanAlbedo, but the northward heat transport between 30°N and 50°N is still weaker by about 0.06 PW and 0.12 PW under G1 and G1oceanAlbedo than under piControl (Fig. 3d). The mitigation of northward heat transport weakening is consistent with the mitigation of AMOC weakening under G1 and G1oceanAlbedo. Northward heat transport weakening between 30°N with 50°N caused by abrupt4xCO$_2$ is more balanced under G1 than with G1oceanAlbedo, consistent with their relative AMOC performance.

Both G4 and G4cdnc significantly mitigate the reduction of northward heat transport between 30°N and 50°N in the North Atlantic basin under RCP4.5 (Fig. 3e). Compared with the RCP4.5 scenario, the G4 and G4cdnc scenarios increase the northward heat transport by about 0.1 PW and 0.08 PW between 30°N and 50°N. The mitigation of northward heat transport weakening between 30°N and 50°N is stronger under G4 than G4cdnc, although differences between G4 and G4cdnc are generally not significant at the 95% level. Change in northward heat transport are thus more complex than their AMOC responses summarized in Table 2.

## 4 Drivers of changes in AMOC

Three drivers of AMOC intensity change have been proposed: i) wind stress at monthly to seasonal periods (Zhao and Johns, 2014); and at annual and decadal scales, ii) changes in seawater density due to varying freshwater flux; and also iii) changes in ocean-air heat exchange (Smeed et al., 2018). We consider each of these in relation to the different SRM experiments. We also look at how model-dependent the drivers are.





## 4.1 Near surface Wind Speed

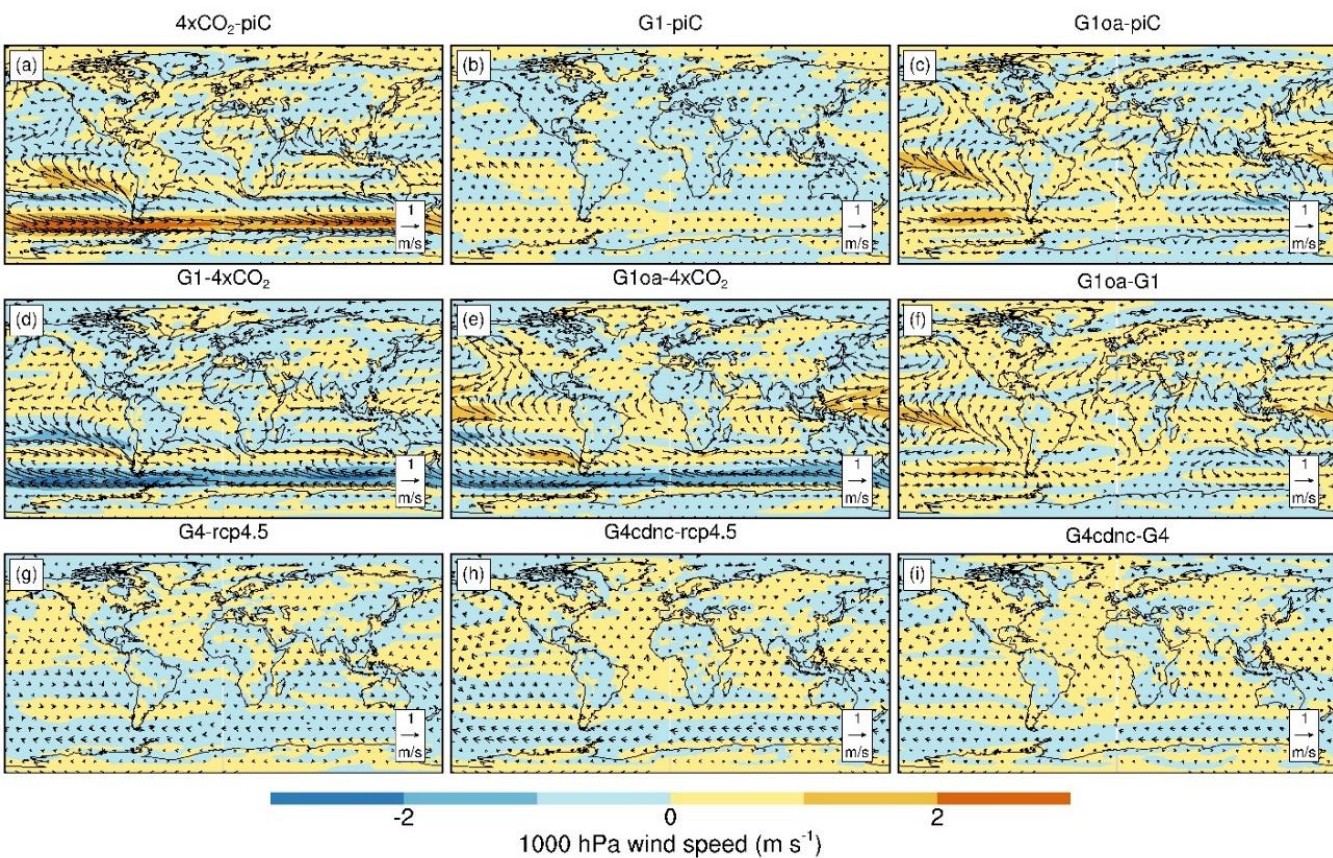

**Figure 4: Spatial distribution of 6 ESM ensemble mean 1000 hPa wind speed and wind direction (arrows) changes under different scenarios (11-50 yr). Blue colors indicate decreased wind speed, the length of arrow in each panel's bottom right represents speeds of 1 m s⁻¹.**





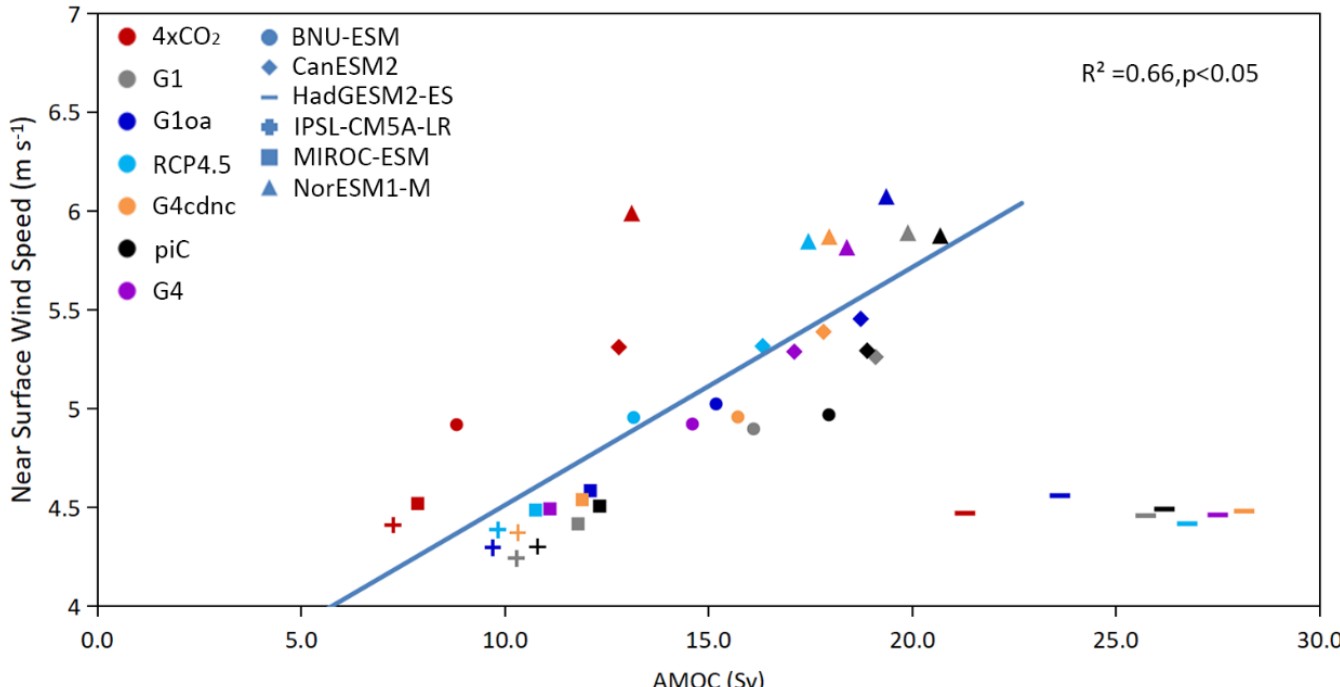

**Figure 5: ESM mean of Wind speed (m s$^{-1}$) over the 40-year analysis period in the whole North Atlantic (North of 30°S). All ESMs except HadGEM2-ES show a high correlation between near surface wind speed and AMOC intensity. The dotted line is the linear regression line of AMOC intensity and wind speed (area average of the whole North Atlantic) over the 40-year analysis period in the 5 ESMs excluding HadGEM2-ES.**

We used the 6 ESMs to calculate near surface wind speed and wind direction under different scenarios. Under the abrupt4×$CO_2$ scenario, the global wind speed has obvious changes compared with other scenarios, especially in the Southern Ocean subpolar westerlies (Fig. 4a). But there is no significant change of wind speed under other scenarios, especially in the Atlantic high latitudes.

There is a significant correlation between wind and AMOC when all models and scenarios except for HadGEM2-ES are selected (Fig. 5). AMOC intensity is significantly related to wind speed within the same scenario, as clearly shown for abrupt4×$CO_2$ in red on Fig. 5, which lies on a relation parallel to, but above, the other scenarios. Similarly, for G1 and piControl points lie on a relation parallel to, but lower, than the mean regression. This suggests that the wind speed is dependent on scenario as well as AMOC, and a direct causal relation between wind and AMOC is not evident. This is consistent with the observation that while wind stress clearly affects AMOC on short timescales it is not the main factor affecting AMOC intensity one long time scales (Zhao and Johns, 2014).





## 4.2 Upward heat flux

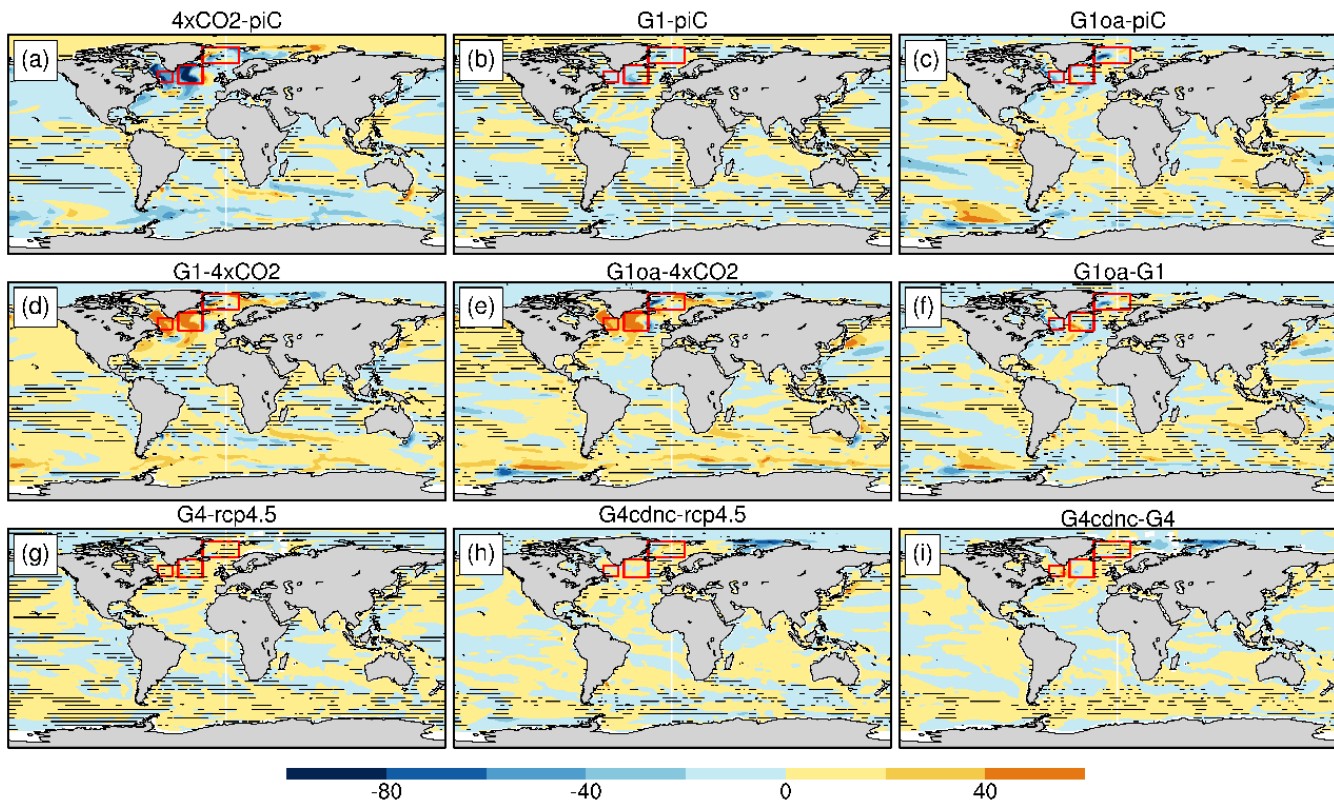


**Figure 6: Upward heat flux change (W m⁻²) in different scenarios(11-50yr). The red boxes mark the three deep convective regions in the northern North Atlantic (from left to right: Labrador, Irminger, and Norwegian seas, (often referred to as the Greenland-Iceland-Norwegian (GIN) Seas). Yellow to orange colors represent an increase in heat flux from the ocean to the atmosphere. Stippling indicates regions where differences are not significant at the 95% level according to the Wilcoxon signed-rank test.**






**Figure 7: 11 yr running annual means simulated by the 6 ESM, and their ensemble mean, of the upward heat flux (W m$^{-2}$) over the 40-year analysis period in the three deep convective regions outlined in Fig.10 under (a) piControl, (b) abrupt4×CO$_2$ and (c) RCP4.5. Panels (d-f) show AMOC anomalies and panels (g-i) the changes relative to the other scenarios: Left column (d,g) relative to piControl; Middle (e,h) relative to global warming scenarios; Right (f,i) relative to other geoengineering scenarios. Colored bands**
**in panels (d-i) represent the across-ESM spread. G1oa refers to G1oceanAlbedo, and PiC refers to piControl.**





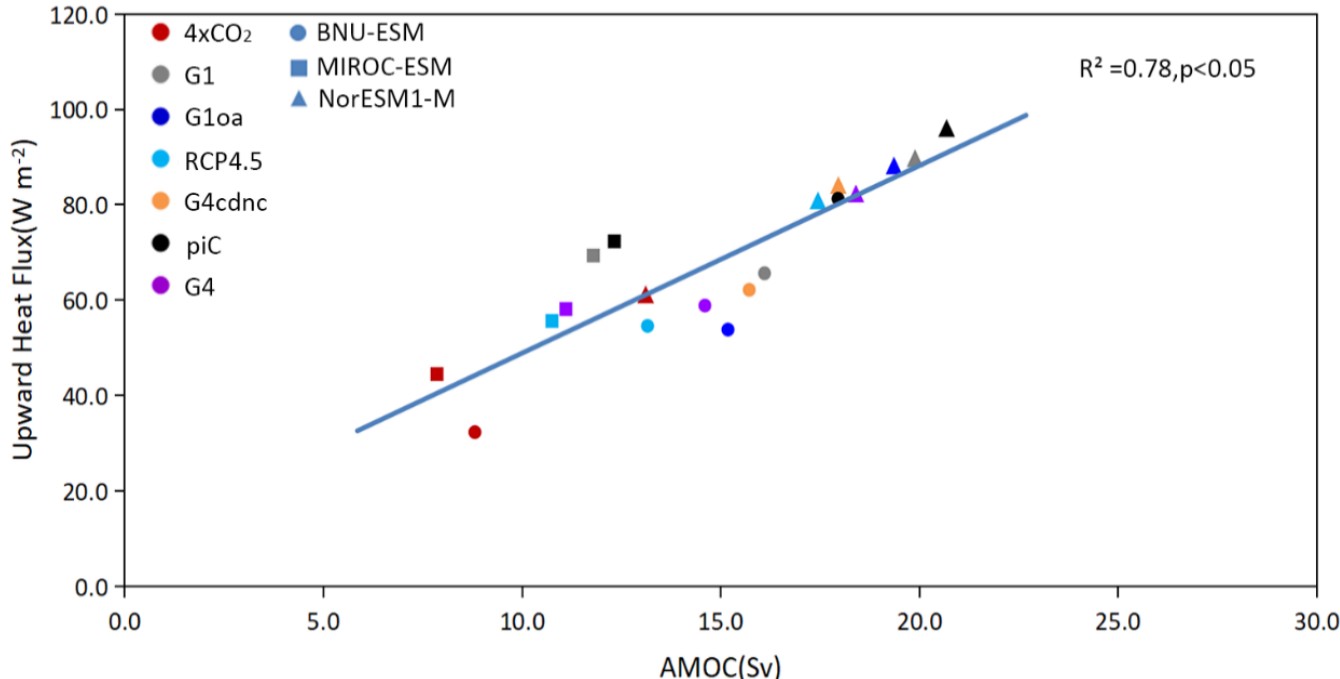

**Figure 8: Model mean of upward heat flux (W m$^{-2}$) over the 40-year analysis period in the three deep convective regions outlined in Fig. 6. Only heat flux data from BNU-ESM, MIROC-ESM and NorESM1-M are available. All the 3 ESMs show a high correlation between upward heat flux (area average of the deep convective regions) and AMOC intensity. The dotted line is the linear regression**
**trendline of AMOC intensity and upward heat flux (area average of the deep convective regions) over the 40-year analysis period.**

AMOC transports heat from the tropics to high latitudes, releases heat in the deep convective regions of North Atlantic, then the surface water density increases and sinks to form cold water flowing southward. Under abrupt4×CO$_2$ and RCP4.5 scenarios, GHG forcing reduces the temperature difference between ocean and atmosphere at high latitudes (Fig. 6 b,c), resulting in reduced heat transfer from the ocean to the atmosphere in the deep convective regions (Fig. 7d). The reduction of upward heat

flux impedes the release of heat from the sea water, weakening the densification process and the rate of sinking in the deep convective region thus weakening AMOC.

Under G1 and G1oceanAlbdedo, the upward heat flux is increased in the deep convective regions relative to abrupt4×CO$_2$, but it is not as high as under the piControl scenario. The changes of upward heat flux in the deep convective region are consistent

with those changes in AMOC intensity. AMOC intensity shows a significant correlation with upward heat flux in the three deep convective regions (Fig. 8), although only 3 models have data fields available. These results show that the change in upward heat flux caused by the modified ocean-atmosphere temperature difference is an important contributor in all ESM to the change in AMOC intensity in the geoengineering scenarios.





## 4.3 Fresh water flux (Precipitation - Evaporation)

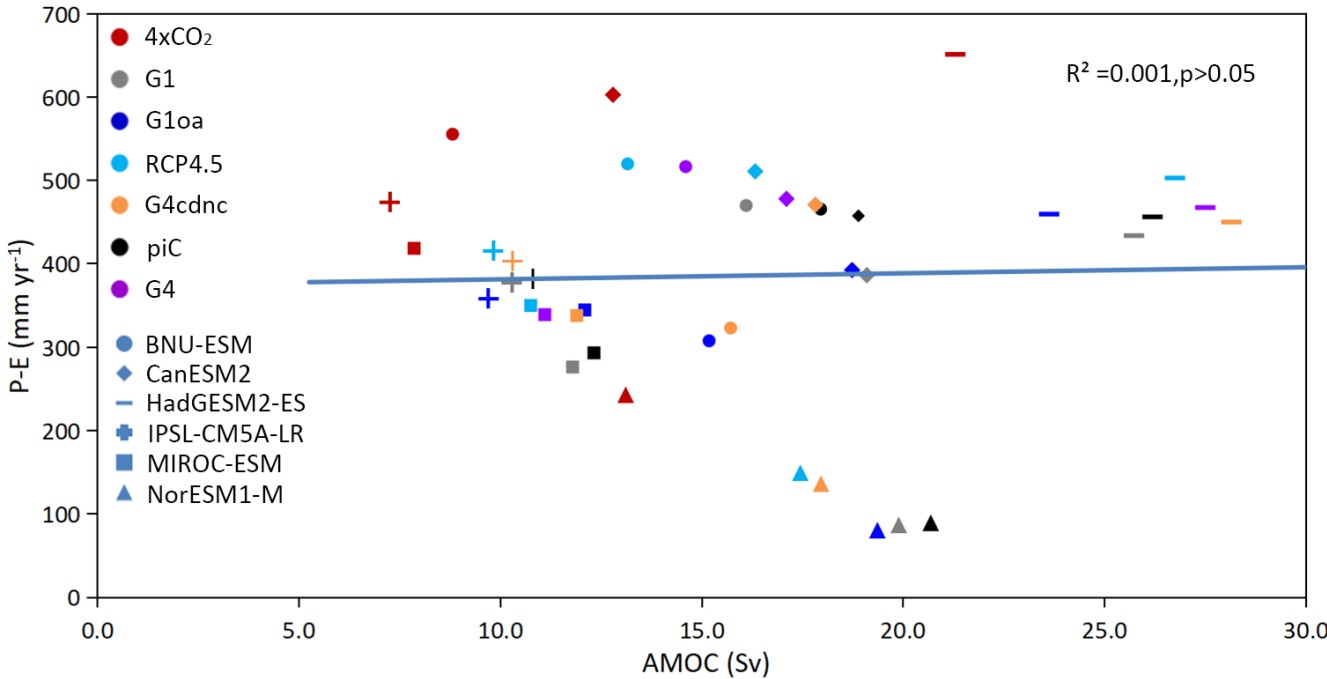


**Figure 9: Model mean of P-E (mm yr⁻¹) over the 40-year analysis period in the three deep convective regions (outlined in Fig. 6). The dotted line is the linear regression trendline of AMOC intensity and upward heat flux (area average of the deep convective regions) over the 40-year analysis period in all 6 ESMs.**

In the output data from ESM, the total fresh water flux into North Atlantic includes precipitation, evaporation, runoff from

river and fresh water flux caused by sea ice thermal dynamic change. Due to the lack of data of runoff and fresh water fluxes caused by sea ice and melting of the Greenland ice sheet, we separately analyze the impact of fresh water flux change caused by precipitation minus evaporation (P-E) on AMOC (Shu et al., 2017).

Relative to piControl, P-E increases by 134 mm yr⁻¹ and 51 mm yr⁻¹ under abrupt4×$CO_2$ and RCP4.5 scenarios respectively.

P-E under the four geoengineering scenarios we studied are decreased compared with their reference GHG forcing scenarios. Geoengineering methods mitigate the increase of P-E under GHG forcing. AMOC intensity has no significant correlation with freshwater flux in the three deep convective regions (Fig. 9), so P-E is not the main driver of AMOC change under the four geoengineering scenarios we studied.




## 4.4 September sea ice extent

**Figure 10: 6 ESMs ensemble mean Arctic minimum sea ice fraction percentage changes (defined as the limit of the 15% ice concentration region) in September in different scenarios (11-50yr). Stippling indicates regions where differences are not significant at the 95% level according to the Wilcoxon signed-rank test.**




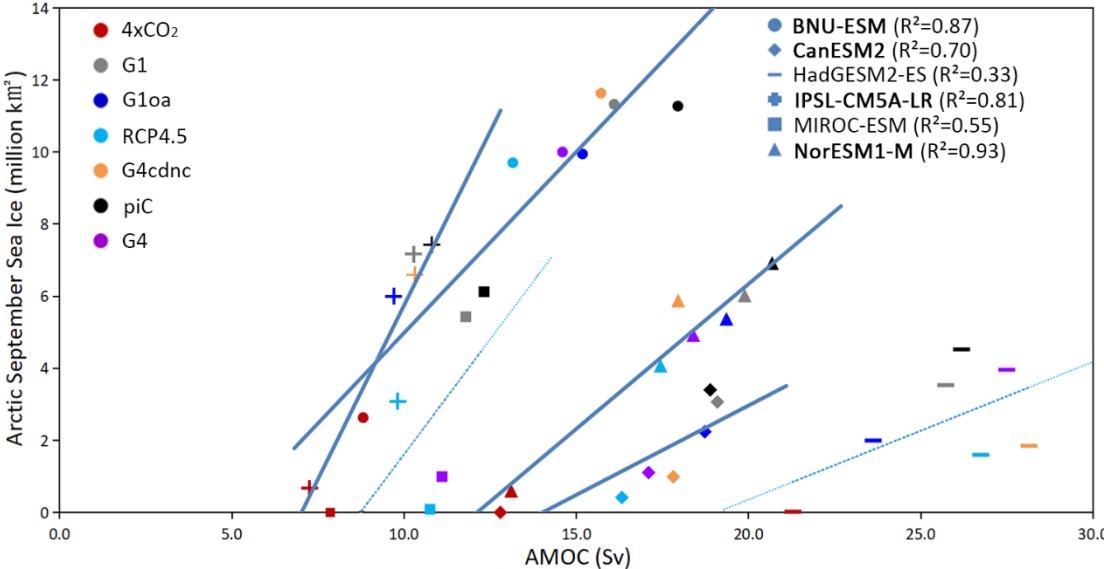

**Figure 11: Model mean Arctic September sea ice area (million km$^2$) over the 40-year analysis period (defined as the limit of 15% ice concentration region). The lines are the linear regression trendlines of AMOC intensity and Arctic September Sea ice area over the 40-year analysis period for each ESM. Those significant at the 95% level are shown as heavier lines and the ESM are labeled in bold in the legend.**

Arctic surface temperatures have risen 2-3 times faster than the global average level leading to loss of Arctic sea ice (Dai et el., 2019). Stronger seasonal sea ice melting weakens the AMOC intensity by injecting fresh water and increasing its storage in the North Atlantic (Li and Fedorov, 2021).  But sea ice cover also reduces heat transport from the sea to the air and hence inhibits ocean convection, which may weaken AMOC (Drijfhout et al., 2012). Therefore, we can analyze the relationship between AMOC intensity and the fresh water flux changes caused by Arctic sea ice thermodynamics via Arctic sea ice extent changes under the four geoengineering scenarios.

There is almost no September sea ice in the northern North Atlantic in any scenario (Fig. 10). The sea ice extent changes most in the northern seas along the Eurasian Arctic coast. Under both abrupt4×CO$_2$ and RCP4.5 scenarios, Arctic sea ice area is greatly reduced (Fig. 10 b, c).

Under the G1 scenario, the spatial pattern of September sea ice is changed relative to piControl, with extent changes of up to 20% regionally, and in total reduced by 3% in an 8 models ensemble (Moore et al., 2014). In our study, relative to piControl, the ensemble mean of 6 ESM sea ice area is decreased by about 9% under G1 and 25% G1oceanAlbedo, which is consistent with their AMOC changes. Indeed, the sea ice of the Norwegian Sea is slightly increased under G1 and G1oceanAlbedo relative to piControl (Fig. 10 b, c). Compared with abrupt4×CO$_2$, September sea ice increases significantly under G1 by about 80%, and by about 64% by G1oceanAlbedo. G1oceanAlbedo produces about 16% less sea ice area than G1 (Table 2). This shows

that G1oceanAlbedo and G1 can mitigate the reduction of Arctic September sea ice caused by GHG forcing, and the mitigation effect under G1 is stronger than G1oceanAlbedo.

Under G4 and G4cdnc, the September sea ice significantly increased by about 22% and 28% compared with RCP4.5. G4cdnc
generally increases sea ice area by 6% more than G4 (Table 2), consistent with AMOC behavior under the two scenarios. All six ESMs agree that the mitigation on the reduction of Arctic September sea ice area under solar dimming is stronger than with MCB and SAI. Mitigation of sea ice loss under G1oceanAlbedo is stronger than G4cdnc, which is all consistent with AMOC behavior under the four scenarios.

To examine the dependence of sea ice extent on AMOC we plot the 40 year mean values for all models and all scenarios (Fig. 11). Such plots can determine how linear are relations, and their across-model differences. In four of the ESM, AMOC is significantly correlated ($p<0.05$) with Arctic sea ice area, HadGem2-ES and MIROC-ESM being the ESM without significant relationship. Although most ESM do have significant relationships between minimum sea ice extent and AMOC strength, the models themselves show large differences in strength of the response, in part due to their AMOC strength in the control
simulations, but also in the changes in September ice extent across the scenarios as seen by the slope of the regression lines in Fig. 11.

There is no significant correlation between AMOC and P-E, but significant correlation between AMOC and Arctic September sea ice area. This also shows that the fresh water changes caused by Arctic September sea ice is the main factor of AMOC
changes under the four Geoengineering. The slopes of the regression lines in Fig. 10 are positive, meaning that greater AMOC strength is correlated with greater ice extent. This contradicts the expected relationship if AMOC were driving sea ice extent since AMOC transports heat to the Arctic. Instead, we observe more sea ice with increased AMOC suggesting that sea ice is instead driving changes in AMOC through the change in fresh water budget.

## 5 Discussion

The mitigation of AMOC intensity, northward heat flux and sea ice extent changes caused by global warming under G1 is significantly stronger than under G1oceanAlbedo. The mitigation under G4cdnc is slightly stronger than G4. The radiative fluxes of the abrupt4xCO$_2$ experiments are 7-8 times greater than those under the RCP4.5 scenarios, as are changes induced in mean and extreme temperatures (Ji et al., 2018). The relative change in AMOC under G4 compared with G1 relative to appropriate controls is similar at about 15%. This compares with about 25% effectivity for G4cdnc relative to G1 and 33%
relative to G1oceanAlbedo (Table S2).  Heat content effectivity for G4 relative to G1 are around 25%, and for G4cdnc the ensemble effectiveness is over 40% of G1oceanAlbedo (Tables S3). The changes in September sea ice extent effectiveness under G4 are about 30% of those under G1 and 50% for G4cdnc relative to G1oceanAlbedo. These comparisons of relative



effectiveness suggest that while MCB is not as effective at ameliorating GHG changes in ocean heat flux, AMOC and hence sea ice extent than the global SRM measures, the specific MCB measures simulated to counteract RCP4.5 are relatively more effective than those under G4. This might mean that specific measures under G4cdnc appear more effective than those simulated under G4 stratospheric aerosol injection, but the forcing applied under G4cdnc was not specifically designed to match the net radiative forcing of the G4 SAI.

We want to examine the differences in response to type of SRM as defined in the GeoMIP experiments we analyze. Because of the large differences in forcing magnitude between, for example Abrupt4xCO$_2$ and RCP4.5 we cannot simply look at anomalies, but instead can compare the responses as a ratio, for example:

$$(G4 - RCP4.5)/(G1 - Abrupt4xCO2) , \quad (3)$$

compares the SAI and the solar dimming anomalies. The ESM have different sensitivities to climate forcing, and to avoid ESM with largest sensitivities dominating an ensemble mean, we further normalize the model fields with a measure of the model sensitivity to climate forcing, which we choose as the top of atmosphere radiative forcing (TOA). So, the measure of efficacy in the example above becomes:

$$\frac{\{(G4-RCP4.5)/(G1-Abrupt4xCO2)\}_{AMOC}}{\{(G4-RCP4.5)/(G1-Abrupt4xCO2)\}_{TOA}} , \quad (4)$$

Which we can calculate for upward heat flux and September sea ice extent in addition to AMOC, and for ratios indicative of the relative responses of MCB to solar dimming and SAI to MCB. The ensemble mean shows sensitivity differences between the ESM, and the ensemble means indicate the typical differences in efficacy between type of geoengineering (Table 3).

**Table 3: Across ESM ensemble mean relative efficacy ratios of geoengineering compared with their efficacy in changing TOA radiation. Where individual ESM have no data, the ensemble mean was used.**

| Type | Ratios | AMOC | Upward Heat Flux | Arctic September Sea Ice |
|---|---|---|---|---|
| SAI/Solar | $\frac{G4 - RCP4.5}{G1 - 4XCO2}$ | 0.6 | 0.4 | 1.0 |
| MCB/Solar | $\frac{G4cdnc - RCP4.5}{G1 - 4XCO2}$ | 0.6 | 0.4 | 0.5 |
| SAI/MCB | $\frac{G4 - RCP4.5}{G4cdnc - RCP4.5}$ | 1.3 | 0.9 | 5.3 |

Table 3 shows that changes to AMOC and upward heat flux are less than for overall climate sensitivity measured as TOA since the ratios are all less than 1. The relative efficacy of by SAI and MCB for AMOC and upward heat fluxes are about half those of solar dimming. Comparing MCB and SAI shows smaller differences, with relative efficacies closer to unity. Arctic September sea ice extent indicates larger differences between type of geoengineering, with SAI being more effective than MCB in the experiments analysed here. Different ESMs have different responses to MCB and SAI, especially the comparison between G4cdnc and G4. Individual model results are shown in Tables S6-8.



Five out six ESM agree SAI is more effective than MCB for AMOC (Table S6), the outlier being HadGEM2-ES. This model also is the only one with greater AMOC intensity under the RCP4.5 and G4 scenarios than in piControl. HadGEM2-ES also is unique in displaying no correlation between wind speed and AMOC (Fig. 5), and along with MIROC-ESM shows an insignificant relation between AMOC and September sea ice extent (Fig. 11).

There is lower consensus on the relative effectiveness with the ESM split three against three for SAI/solar and MCB/solar. In the case of upward heat flux (Table S7), the ESM generally agree that solar dimming is more effective than either SAI or MCB with little to choose between SAI and MCB. For September sea ice, (Table S8). SAI clearly outscores MCB in all but BNU-ESM of the models and experiments we analyzed, while SAI and solar are fairly similar in effectiveness across the ESM.

The proximate factor from our analysis of the main drivers of changes under the different scenarios, is the change in heat flux transported from ocean to atmosphere caused by the air-sea temperature difference changes in deep convective regions of the North Atlantic (Fig. 7). This is consistent with the analysis of the G1 experiment (Hong et al., 2017). All the geoengineering scenarios produce surface cooling, which partially restores the ocean-atmosphere temperature contrast altered by GHG forcing, and thus increases the heat flux from ocean to atmosphere. In the three deep convection regions of the northern North Atlantic

(Figure 6), the ocean temperature is usually higher than the near surface temperature, and the surface seawater originating from the tropics can release heat to the atmosphere and cool down. Global warming increases the near-surface air temperature, reducing the air-sea heat exchange. While geoengineering might be expected to ameliorate this problem by cooling the atmosphere, we found that the surface air temperatures in the deep convection regions of the North Atlantic remained higher than in piControl, even in scenarios in which the global radiative budget is balanced. Thus, the geoengineering scenarios had

an intermediate amount of AMOC weakening, in between the pre-industrial state and unmitigated global warming. The decrease of surface seawater density in the northern North Atlantic caused by the increase of surface seawater temperature weakens the surface seawater subsidence in this area under the four SRM geoengineering scenarios as compared with piControl.

While changes in upward heat flux over the three convective regions play a prominent role, the fresh water flux changes caused

by Arctic sea ice melting also affects AMOC changes under geoengineering. Sea ice melting releases large amounts of fresh water into the North Atlantic. In the deep convection regions of the North Atlantic, the injection of a large amount of fresh water reduces the density of surface sea water, hindering surface water sinking, and weakening AMOC. Although the wintertime formation of Arctic sea ice increases the density of the surface water in the Arctic, promoting surface water sinking in the deep convection regions of the North Atlantic, the sustained decline in Arctic sea ice and strengthened seasonal cycle

produces a gradual freshening of the upper Arctic Ocean (Li and Federov, 2021). The four geoengineering experiments thus can mitigate the AMOC weakening caused by GHG forcing through increasing the September Arctic sea ice area and reducing sea ice seasonality. The changes of Arctic sea ice area and AMOC are interactive, and the changes of Arctic September sea

ice area are significantly correlated under different scenarios (except for HadGEM2-ES). However, the response is ESM-dependent as the relationship between AMOC changes and Arctic sea ice area changes are different. A key uncertainty when

it comes to the AMOC in the future is the melting of the Greenland ice sheet. More realistic modelling of the ice sheet or sensitivity studies (e.g. Swingedouw et al. 2015), or indeed interactive ice sheet modelling would be needed to address this, which is beyond the scope of this study.

Change of near surface wind speed are known to alter the speed of northward surface water transport, and thus AMOC. The

effects of wind speed appear on short time scales (Yang et al., 2016). Near surface wind speed changes over North Atlantic is correlated with AMOC under many of the different scenarios, but they are not significantly correlated under across scenarios simulated by each ESM. Thus, scenario impacts windspeed as expected, but a consistent relation between scenarios simulated by each ESM is not evident. Hence near surface wind speed is not the main factor of AMOC changes under different scenarios.

## 6 Conclusions

GHG forcing weakens AMOC intensity reducing northward ocean heat transport. The three SRM methods we studied, solar dimming (G1), MCB (G1oceanAlbedo and G4cdnc) and SAI (G4) mitigate the AMOC weakening caused by GHG forcing. The mitigation effect of AMOC weakening under MCB are similar as SAI, but both are relatively less effective than solar dimming in these experiments. All the four geoengineering scenarios demonstrate weakened AMOC compared with the piControl scenario. The drivers producing the changes in AMOC are dominated by the differences in surface air-ocean

temperatures, with the radiative cooling produced by the SRM tending to reverse the GHG changes. We found no relationship between freshwater flux due to river flow or imbalance in precipitation-evaporation and changes in AMOC, but there is a significant correlation between September sea ice extent and AMOC intensity. The bigger the decline in sea ice extent, stronger the reduction in AMOC intensity. This suggests that AMOC is not driving sea ice reduction since a lower AMOC means less ocean heat transport, but it does indicate that freshening mechanisms in the deep convection regions associated with greater

sea ice seasonality act to reduce AMOC as summer sea ice is removed.

## 7 Data availability

All data used in this study, except for part of the data of NorESM1-M and IPSL-CM5A-LR, are available through the Earth System Grid Federation (ESGF) Network (http://esgf.llnl.gov). The NorESM1-M and IPSL-CM5A-LR data are archived by the modeling team.




*Author contributions.* John C. Moore and Mengdie Xie conceived and designed the analysis. Mengdie Xie mainly collected all the data and performed the analysis. Mengdie Xie wrote the paper, and John C. Moore, Liyun Zhao, Michael Wolovick and Helene Muri provided critical suggestions and revised the paper. All authors contributed to the discussion.

*Competing interests.* The authors declare that they have no conflict of interest.

## Acknowledgments

This study is supported by the National Key Research and Development Program of China (2018YFC1406104), National Natural Science Foundation of China (No. 41941006), National Basic Research Program of China (2016YFA0602701), Finnish Academy COLD consortium grant 322430 and Uninett Sigma2 resources NS9033K. We thank two anonymous

reviewers made many suggestions to help improving the paper.

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
