# Peer review of "Impacts of three types of solar geoengineering on the Atlantic Meridional Overturning Circulation"

_Atmospheric Chemistry and Physics, 2021_

## Author Response (AR1)

**Reply to Referee's Comments**

Dear reviewers,

Re: Manuscript ID: acp-2021-877 and Title: Impacts of three types of solar geoengineering on the North Atlantic Meridional Overturning Circulation.

Thank you for your comments concerning our manuscript. We have studied your comments carefully and have made corrections. Revised parts are marked in red in the manuscript. The main corrections in the paper and the responses to your comments are as following:

**Responds to the comments of Reviewer #1:**

**General Comment:** The authors study the efficacy of different geoengineering on ameliorating the AMOC reduction under GHGs forcing using ESM simulations. While I suspect the author's analyses were constrained by what's available in the GeoMIP output, could you explain why G1 and G1oa were used to counter 4xCO2 forcing whereas G4 and G4cdnc were to counter RCP4.5 scenario? The authors are fully aware that GHG forcing in 4xCO2 and RCP4.5 is very different, and the geoengineering forcing strength is also different between G1, G1oa, G4, G4cdnc. These differences render the comparison across G1/G1oa and G4/G4cdnc somewhat arbitrary, and this is true whether you are talking about an absolute anomaly (e.g., table 2), or a ratio (as in equation 3), or ratio's ratio (as in equation 4). But, if it has to be done this way, you should provide more justification and/or motivation. Alternatively, you can compare G1 with G1oa, and G4 with G4cdnc without the cross-group comparisons. The presentation is otherwise generally clear, except for a few places (see specific comments below).

**Reply:** We have no choice in the selection of control greenhouse gas (GHG) scenarios as they are designed with specific GHG forcing. This is because that G1 and G1oa are designed to completely offset the global mean radiative forcing due to a $CO_2$ quadrupling experiment (abrupt4×CO2), while in GeoMIP experiment G4 and G4cdnc, the radiative forcing due to the representative concentration pathway 4.5 (RCP4.5) scenario are partly offset. These experiments are explained in the following table, which we add in the text. The comparisons between G1oa and G1 separately, and G4 and G4cdnc are done throughout the paper, but this is a little elementary. Taking the ratios is the only way of compensating for the differing signal strengths, as the applied forcing is much larger in the abrupt4xCO2 scenario and the geoengineering scenarios associated with it (G1 and G1oa) than in the RCP4.5 scenario and its associated geoengineering scenarios (G4 and G4cdnc). Taking the ratio of ratios then allows us to compare across the differing geoengineering methods, which is really the point of the paper. Ratios are a standard way of normalizing results with different forcing, for example in medicine: Curran-Everett 2013 Adv Physiol Educ. 37(3):213-9. https://doi.org/10.1152/advan.00053.2013). The only arbitrary choice here is the selection of TOA radiation as the metric used to quantify the

strength of the forcing, another choice might have been mean global mean temperature, but we think that TOA radiation was a little more fundamental since it is used in the definition of model climate sensitivity.

The following has been added to the manuscript to clarify these choices:

Table 2: A summary of the four geoengineering experiments included in the analysis.

| Scenario | Background | Objective | Geoengineering Type |
|---|---|---|---|
| G1 | Abrupt 4×CO2 | radiative balance | Solar Dimming (SD) |
| G1oceanAlbedo | Abrupt 4×CO2 | radiative balance | Idealized Marine Cloud Brightening |
| G4 | RCP4.5 | radiative offset | Stratospheric Aerosol Injection (SAI) |
| G4cdnc | RCP4.5 | radiative offset | Marine Cloud Brightening (MCB) |

In the following analysis, we make comparisons between G1oa and G1, and G4 and G4cdnc separately as they do not use the same greenhouse gas forcing backgrounds (Table 2). But we are also interested in comparing the different geoengineering types and doing this can be done with the ratios of their response, e.g. $(G4 - RCP4.5)/(G1 - Abrupt4xCO2)$. The different ESM also have different climate sensitivities, and we also account for this by considering their top of atmosphere radiative forcing (TOA).

**Specific Comments**

**Comment No.1:** Line 187-188, "Generally, mitigation of AMOC weakening under G4cdnc is more than with G4, but weaker than G1 solar dimming": But mitigation of G1 solar diming was applied to 4xCO2 not RCP4.5, so this comparison is not apples-to-apples.

**Reply:** Yes, as we do say in the same sentence immediately following the part the referee quotes: "but these scenarios were not designed to have identical forcing, so we shall discuss their relative efficacy later in the Discussion." This is because the level of forcing applied under the G4 scenarios is weaker than under the G1, as is clear from the new Table 2 shown above. We need to use the normalized ratios to make cross-group comparisons of the efficacy of the different types of geoengineering, and for ranking of these scenarios. The motivation for this cross-group comparison comes from Ji et al., 2018:

Ji, D., Fang, S., Curry, C. L., Kashimura, H., Watanabe, S., Cole, J., Lenton, A., Muri, H., Kravitz, B., and Moore, J. C.: Extreme temperature and precipitation response to solar dimming and stratospheric aerosol geoengineering, Atmos. Chem. Phys., 18, 10133–10156, doi: 10.5194/acp-18-10133-2018, 2018.

**Comment No.2:** Fig. 4, difference plot: Is there a reason why you didn't perform the statistical significance test here?

**Reply:** The figure of wind speed and wind direction now shows significance (the new Fig. 4), and the supplemental wind speed figures show each model individually (the Fig. S1 in the Supplementary Information). We do note in the text: "Under the abrupt4×CO2 scenario, the global wind speed has obvious changes compared with other scenarios, especially in the Southern Ocean subpolar westerlies (Fig. 4a). But there is

no significant change of wind speed under other scenarios in the North Atlantic high latitudes."

[Figure]

**Figure 4: Spatial distribution of 6 ESM ensemble mean 1000 hPa wind speed and wind direction (arrows) changes under different scenarios (11-50 yr). Blue colors indicate decreased wind speed, the length of arrow in each panel's bottom right represents speeds of 1 m s⁻¹. Translucent white overlay indicates regions where differences are not significant at the 95% level according to the Wilcoxon signed‐rank test.**

**Comment No.3:** Line 233, Fig. 5 caption, "in the whole North Atlantic (North of 30°S)": Within this large domain, wind in the subpolar NA (e.g., north of 45n) in particular may matter more than wind in the other regions. Have you done a similar calculation but use wind in the NA?

**Reply:** We have done a similar analysis for the North Atlantic north of 45°N. The following are now in the supplementary information:

Similar results were obtained for winds only over the deep convection regions, and for just the Atlantic north of 45°N (Fig. S2).

[Figure]

**Figure S2: AMOC intensity (Sv) versus near-surface wind speed (m s⁻¹), where the different colors indicate the different experiments, and the shapes represent the six different ESMs. All ESMs except HadGEM2-ES show a high correlation between the near-surface wind speed and AMOC intensity. The dotted line is the linear regression line of AMOC intensity and wind speed (area average of the subpolar North Atlantic) over the 40-year analysis period in the 5 ESMs excluding HadGEM2-ES.**

**Comment No.4:** Line 245, "…is dependent on": Change it to something like "is correlated with", so no causality is implied.
**Reply:** Done.

**Comment No.5:** Line 246, "and a direct causal relation between wind and AMOC is not evident": Change it to something like "but this analysis does not address causal relation between wind and AMOC."
**Reply:** Done.

**Comment No.6:** Line 339-340, "This also shows that the fresh water changes caused by Arctic September sea ice is the main factor of AMOC 340 changes under the four Geoengineering.": Please clarify. What about the heat flux you just described? Is it not a main factor?
**Reply:** Yes, it is. The fresh water flux changes caused by Arctic sea ice melting is part of the heat flux change mechanism rather than e.g. P-E or wind. But the proximate factor from our analysis of the main drivers of changes under the different scenarios, is the change in heat flux transported from ocean to atmosphere. So, we changed the sentence to be "fresh water changes caused by Arctic September sea ice is a key factor in AMOC changes under the four geoengineering experiments."

**Comment No.7:** Line 354-356, "the specific MCB measures simulated to counteract RCP4.5 are relatively more effective than those under G4. This might mean that specific measures under G4cdnc appear more effective than those simulated under G4

stratospheric aerosol injection,": If I read it correctly, the second sentence largely repeats the first sentence, right? Please clarify.

**Reply:** Yes, we deleted the first sentence. So it now reads: "The changes in September sea ice extent effectiveness under G4 are about 30% of those under G1 and 50% for G4cdnc relative to G1oceanAlbedo. This might mean that specific measures under G4cdnc appear more effective than those simulated under G4 stratospheric aerosol injection, but the forcing applied under G4cdnc was not specifically designed to match the net radiative forcing of the G4 SAI."

**Comment No.8:** "but the forcing applied under G4cdnc was not specifically designed to match the net radiative forcing of the G4 SAI.": Precisely. So what does the comparison tell you?

**Reply:** This comparison shows that G4cdnc is more effective than G4 in mitigating ocean heat flux, AMOC and sea ice extent. But the relative efficacy is more important and that is discussed in the next paragraph. Explaining why we need more complex analysis of ratios is the reason for the sentence. To make this even clearer, we added the following:

but the forcing applied under G4cdnc was not specifically designed to match the net radiative forcing of the G4 SAI.

Hence, we also investigate the relative efficacies.

**Comment No.9:** Line 360-361, "we cannot simply look at anomalies, but instead can compare the responses as a ratio,": Ratio is not less arbitrary than anomalies. Is there a reason why G1 was not done to counter RCP4.5 as well like G4 was?

**Reply**: Normalizing responses with ratios is a very common way of standardizing experimental responses to different forcings, see e.g. Ji et al., 2018 in this specific field and in medicine e.g. Curran-Everett 2013 Adv Physiol Educ. 37(3):213-9. https://doi.org/10.1152/advan.00053.2013). This is a simple way of comparing the experiments because, as the new Table 2 makes clear the experimental design of the ESM simulations dictated the GHG forcing used. This is because G1 and G1oa are designed to completely offset the global mean radiative forcing due to a CO2-quadrupling experiment (abrupt4×CO2), while in GeoMIP experiment G4 and G4cdnc, the radiative forcing due to the RCP4.5 scenario are only partly offset. And we are bound by the experiments and simulations available from GeoMIP.

**Comment No.10:** Line 12, cross out "North" before "Atlantic Meridional…"
**Reply:** Done.

**Comment No.11:** Line 60, cross out "side" before "effects that SRM…"
**Reply:** Done.

**Comment No.12:** Line 174 -175, "differences which are significant at the 95% level.": Table 2 shows 1.4 sv is significant, 0.7 sv is not.
**Reply:** Thanks! Wording changed to "The average AMOC intensity is insignificantly

weaker under G1, but statistically significant lower by 1.4 Sv under G1oceanAlbedo (p<0.05; Table 3) than under piControl".

**Comment No.13:** Line 183 -184, "The difference between G4cdnc with G4 over the 40-year analysis period is also significant": Table 2 shows that this difference is 0.6 Sv which is not statistically significant. Could you clarify?
**Reply:** Yes, thanks for spotting this mistake which is now deleted.

*Response to the comments of Reviewer #2:*

**General Comments:** The study examined the relationship between the AMOC responses under radiatively forced experiments and the geoengineering experiments used to mitigate the warm. The paper is overall well written and clearly presented, and the conclusion of the efficacy of the mitigation geoengineering method logical.

However, my major comment is about the mechanism proposed to explain the AMOC response differences in the experiments, that is the sea ice-driven response. The main evidence used to support this inference is the mainly correlation between AMOC strength and Sea ice extent. They argue that the correlation should be negative if the sea ice extent is caused by the AMOC, but the correlation found here is positive. The expected negative AMOC-sea ice extent correlation is based on the assumption that an increase in the AMOC should transport more heat into the Arctic and thus reduce sea ice extent. However, several studies have shown that that heat transport into the Arctic increases with AMOC weakening under global warming. In fact, this heat transport increase into the Artic is also seen in Figure 3, poleward of 60N and agrees with sea ice extent differences between the experiments. Under this scenario, it could also be argued that a positive correlation AMOC - sea ice extent is caused by the AMOC.

**Reply:** Yes, as the Reviewer pointed out, it can be seen in Figure 3e that the Northward Heat Transport changes sign at about 60° N, and this does require some discussion of our interpretation of the sea ice extent mechanism.

Therefore, we analyzed the correlation between the North Atlantic heat transport across 60° N (0-700m) and the Arctic September sea ice extent (a new Fig. S3). The correlation between the change of North heat transport at 60° N and the Arctic September sea ice extent is not significant. This lack of correlation can be compared with that in Fig. 11 where only HadGem2-ES has a lower $R^2$ than 0.5.

We therefore include the following text: The slopes of the regression lines in Fig. 11 are positive, meaning that greater AMOC strength is correlated with greater ice extent. However, Fig. 3e also shows that heat transport anomalies under the geoengineering scenarios change sign at about 60°N, with reductions in heat transport in the south coinciding with increases to the north of 60°N. But correlations of heat transport across 60°N with sea ice extent for separate ESM across scenarios are all insignificant and vary in sign (Fig. S3), in stark contrast to the regression lines in Fig. 11.

[Figure]

Figure S3. Northward ocean heat transport (PW) versus Arctic September sea ice area (million $km^2$) over the 40-year analysis period (defined as the limit of 15% ice concentration region). The dotted lines are the linear regression trendline of Northward Ocean Heat Transport (0-700m, 60°N) and Arctic September Sea ice area over the 40-year analysis period. The $R^2$ for all data points is an insignificant 0.26.

It is, however, true that the G4 and G4cdnc and the RCP4.5 experiments are significantly correlated (which we now include as fig S4). The reason for the relation is presumably as the referee suggested- the increased heat flux north of 60°.

For individual scenarios, there are significantly anticorrelations only for the RCP4.5, G4 and G4cdnc scenarios (Fig. S4). In this respect, the behaviour is similar, although less robust, as for wind forcing in Fig. 5, where scenario impacts as expected, but a consistent relation between scenarios simulated by each ESM is not present. The stronger sea ice correlation with increased AMOC suggests that sea ice may be driving changes in AMOC through the change in fresh water budget.

[Figure]

Figure S4. Model mean Arctic September sea ice area (million km²) over the 40-year analysis period (defined as the limit of 15% ice concentration region). The lines are the linear regression trendlines of Northward Ocean Heat Transport (0-700m, 60°N) and Arctic September Sea ice area over the 40-year analysis period for each scenario. Those significant at the 95% level are shown as heavier lines in bold in the legend.

**General Comments (continued)** The earlier paper also cited to support this mechanism (Li and Fedorov 2021) is also primary forced by sea ice changes rather than the radiative forcing in the experiments in this study, so the conclusions from this study do not necessarily carry over. The authors should provide more evidence support the causality they're inferring from this study.

**Reply:** Our interpretation of Li and Federov (2021), contrary to the Reviewer, is that their experiments were made with perturbations in radiative forcing, represented as an imposed radiative flux imbalance at the sea ice surface, and that is driving the changes. Here are several quotes from Li and Fedorov (2021) to support this, e.g. in the abstract: "*Here, we examine global ocean salinity response to such changes of Arctic sea ice using simulations wherein we impose a radiative heat imbalance at the sea ice surface*"; in Section 2 "*Sea ice surface radiative balance is altered by either reducing sea ice surface albedo to increase shortwave absorption (named "SW" experiment) or reducing the sea ice surface emissivity to restrain outgoing long wave radiative fluxes (named "LW" experiment)*." and "*Two additional experiments are also conducted with stronger shortwave absorption ("strong-SW") and weaker longwave emission ("weak-LW"). All simulations start from a quasi equilibrium preindustrial control climate. Sea ice perturbations are initiated from the beginning of each simulation and maintained for 200 years. The magnitude of maximum sea ice reduction is roughly proportional to the strength of sea ice radiative perturbations*." Additionally, the experiments shown in their Fig. 1 all seem to be radiative forcing designs. Thus, it does seem to be

conceptually comparable with the radiative forcing differences we examine in our geoengineering experiments.

We argue that the higher significance of correlation between increased AMOC and sea ice area being more significant than heat transport at 60°N supports the view that sea ice is affecting AMOC in the experiments. The crucial question seems to be: can changes in sea ice in the Arctic at least contribute to changes in AMOC? There seems to be support from other authors looking at freshening and dynamic changes in the Arctic Ocean – e.g. Wang et al. (2019, https://doi.org/10.1175/JCLI-D-18-0237.1), "*the changes in the Arctic freshwater spatial distribution indicate that the influence of sea ice decline on the ocean environment is remarkable. Sea ice decline increases the amount of Barents Sea branch AW in the upper Arctic Ocean, thus reducing its supply to the deeper Arctic layers. This study suggests that all the dynamical processes sensitive to sea ice decline should be taken into account when understanding and predicting Arctic changes*".

Li and Federov, 2021 also note that "*While the mechanisms of this ongoing Arctic freshening remain under debate, on multidecadal timescales the low salinity anomalies can potentially escape the Arctic and affect ocean deep convection sites in the subpolar North Atlantic, weakening the Atlantic Meridional Overturning Circulation (Scinocca et al. 2009; Oudar et al. 2017; Sevellec et al. 2017; Liu et al. 2018; Liu and Fedorov 2019*)."

We therefore modify the manuscript text as follows: Wang et al. (2019), note that sea ice decline is likely to have remarkable influence on the ocean environment and that sea ice decline impacts on dynamical processes should be considered. Climate model sensitivity studies perturbing both sea ice and radiative forcing (Sevellec et al. 2017; Liu et al. 2018; Liu and Fedorov 2019) elucidate how buoyancy anomalies may escape the Arctic into ocean deep convection regions weakening the AMOC.

And in the conclusion, we also modify the wording: The strong statistical relationship for most models across scenarios suggests that AMOC is not directly driving sea ice reduction since a lower AMOC means less ocean heat transport. Instead it supports modelling studies that indicate freshening mechanisms in the deep convection regions associated with greater sea ice seasonality may act to reduce AMOC as summer sea ice is removed.

References added:

Liu W., Fedorov A.V.: Global impacts of Arctic sea ice loss mediated by the atlantic meridional overturning circulation. Geophys. Res. Lett., 46, 944–952. doi: 10. 1029/ 2018G L0806 02, 2019.

Liu W., Fedorov A., Sevellec F.: The mechanisms of the Atlantic Meridional Overturning Circulation slowdown induced by Arctic sea ice decline, Journal of

Climate, doi: 10. 1175/ JCLI-D- 18- 0231.1, 2018.Sévellec F., Fedorov A. V., Liu W.: Arctic sea-ice decline weakens the Atlantic Meridional Overturning Circulation. Nature Clim. Change, 7, 604–610, doi: 10. 1038/ nclim ate33 53, 2017.Wang, Q., Wekerle, C., Danilov, S., Sidorenko, D., Koldunov, N., Sein, D., Rabe, B., and Jung, T.: Recent Sea Ice Decline Did Not Significantly Increase the Total Liquid Freshwater Content of the Arctic Ocean, Journal of Climate, 32(1), 15-32, 2019. Retrieved Dec 28, 2021, from https://journals.ametsoc.org/view/journals/clim/32/1/jcli-d-18-0237.1.xml.

**Specific Comments**

**Comment No.1:** Title: Not sure "North" is appropriate before Atlantic Meridional overturning circulation"

**Reply:** Deleted "North" before Atlantic Meridional overturning circulation, as well as in the abstract.

**Comment No.2:** Line 165: Last sentence is not clear "and RCP4.5" probably should be removed.

**Reply:** Done.

---

## Author Response (AR2)

**Reply to Referee's Comments**

Dear reviewers,

Re: Manuscript ID: acp-2021-877 and Title: Impacts of three types of solar geoengineering on the North Atlantic Meridional Overturning Circulation.

Thank you for your comments concerning our manuscript. We have studied your comments carefully and have made corrections. Revised parts are marked in red in the manuscript. The main corrections in the paper and the responses to your comments are as following:

**Suggestions for revision or reasons for rejection (will be published if the paper is accepted for final publication)**

The revised version has improved over the previous submission. My only remaining comment is that eqn 3, eqn 4, and the associated text (on page 21 of the manuscript named "… -version2") can be made much easier to understand if you

1) First define a sensitivity parameter in terms of climate response signals divided/scaled by TOA radiation anomalies, written as P/R, where P is, for example, (G4-RCP4.5)_ AMOC, and R is (G4-RCP4.5)_TOA. This puts different mitigation experiments in equal footing.

2) Then use ratio of P/R between different mitigation experiments to compare efficacy. In math form, this would be (P1/R1)/(P2/R2). While this new form is equivalent to what you wrote in eqn 4 (P1/P2)/(R1/R2), it's easier to interpret and explain.

The text following Line 368 can be changed to something like:

"…. Because of the large differences in forcing magnitude between, for example Abrupt4xCO2 and RCP4.5, we first define climate response sensitivity parameters as :

(G4-RCP4.5)_ AMOC/ (G4-RCP4.5)_TOA,
(G1-4XCO2)_ AMOC/ (G1-4XCO2)_TOA (3)

which are AMOC changes per unit change of the corresponding TOA radiation flux changes. Then we compare the efficacy of different mitigation experiments by the ratio of their sensitivity parameters, e.g.:

$$\frac{(G4\text{-}RCP4.5)\_ AMOC/ (G4\text{-}RCP4.5)\_TOA}{(G1\text{-}4XCO2)\_ AMOC/ (G1\text{-}4XCO2)\_TOA} \quad (4)$$
…."

Then you can go on to describe Table 4 and interpret the results.

Thanks for your suggestions.

We think the use of sensitivity parameter is a good idea which make the equations easier to understand and interpret. But the symbols P, R may make the text more complicated and non-intuitive, because there are three type of experiments and three variables.

The revised parts are marked in red as follows:

We want to examine the differences in response to type of SRM as defined in the GeoMIP experiments we analyze. The ESM have different sensitivities to climate forcing so we normalize the model fields with top of atmosphere radiative forcing (TOA), for example:

$$(G4 - RCP4.5)_{AMOC}/(G4 - RCP4.5)_{TOA}, \tag{3}$$

which are AMOC changes per unit change of the corresponding TOA radiation flux changes.

Because of the large differences in forcing magnitude between, for example Abrupt4xCO$_2$ and RCP4.5 we cannot simply look at anomalies, but instead can compare the responses as a ratio, for example:

$$(G4 - RCP4.5)/(G1 - Abrupt4 \times CO2) , \tag{4}$$

compares the SAI and the solar dimming anomalies.

Then we compare the efficacy of different mitigation experiments by the ratio of their sensitivity parameters, for example the measure of efficacy in the example of comparing the SAI and the solar dimming anomalies above becomes:

$$\frac{(G4 - RCP4.5)_{AMOC}/(G4 - RCP4.5)_{TOA}}{(G1 - Abrupt4 \times CO2)_{AMOC}/(G1 - Abrupt4 \times CO2)_{TOA}}, \tag{5}$$

Which we can calculate for upward heat flux and September sea ice extent in addition to AMOC, and for ratios indicative of the relative responses of MCB to solar dimming and SAI to MCB. The ensemble means indicate the typical differences in efficacy between type of geoengineering (Table 4).